# Eye lens crystallin proteins inhibit the autocatalytic amyloid amplification nature of mature α-synuclein fibrils

**Ricardo Gaspar** [1,2¤]*, **Tommy Garting**[1], **Anna Stradner** [1]*

**1** Department of Physical Chemistry, Lund University, Lund, Sweden, **2** Biochemistry and Structural Biology, Lund University, Lund, Sweden

¤ Current address: International Iberian Nanotechnology Laboratory, Braga, Portugal
* ricardo.gaspar@inl.int (RG); anna.stradner@fkem1.lu.se (AS)

## Abstract

Parkinson´s disease is characterized by the accumulation of proteinaceous aggregates in Lewy bodies and Lewy Neurites. The main component found in such aggregates is α-synuclein. Here, we investigate how bovine eye lens crystallin proteins influence the aggregation kinetics of α-synuclein at mildly acidic pH (5.5) where the underlying aggregation mechanism of this protein is dominated by secondary nucleation of monomers on fibril surface providing an autocatalytic amyloid amplification process. Bovine α-, βH- and γB-crystallins were found to display chaperone-like activity inhibiting α-synuclein aggregation. This effect was shown to be time-dependent, with early additions of α-crystallin capable of retarding and even inhibiting aggregation during the time frame of the experiment. The inhibitory nature of crystallins was further investigated using trap and seed kinetic experiments. We propose crystallins interact with mature α-synuclein fibrils, possibly binding along the surfaces and at fibril free ends, inhibiting both elongation and monomer-dependent secondary nucleation processes in a mechanism that may be generic to some chaperones that prevent the onset of protein misfolding related pathologies.

## Introduction

Protein homeostasis is essential for health and survival of cells. With aging *in vivo* quality control systems become less efficient of maintaining a stable and functional proteome [1]. This gives rise to protein misfolding and aberrant aggregation associated with multiple neurodegenerative disorders, such as, Parkinson´s disease (PD) [2,3]. PD neurodegeneration results from the accumulation of intercellular and intracellular deposits of amyloid aggregates. Both the formation of protein-rich aggregates and spreading of the pathology throughout the brain are hallmarks of the disorder [4]. The main component found in these inclusion bodies is a protein known as α-synuclein (α-syn) [5]. α-Syn is a 140 amino-acid long acidic protein with three distinct regions: an N-terminal lipid-binding region, a hydrophobic central region and a C-terminal acidic tail. *In vivo* α-syn can be found with both an unfolded conformation in the cytosol and with α-helical conformation associated to lipid membranes [6]. Due to its ability

**Funding:** We gratefully acknowledge financial support from the Knut and Alice Wallenberg

Foundation (Project Grant KAW 2014.0052), the Swedish Research Council (VR Grant 2016-03301) and the Faculty of Science at Lund University. The funders had no role in study design, data collection and analysis, decision to publish, or preparation of the manuscript.

**Competing interests:** The authors have declared that no competing interests exist.

to interact with lipid membranes and location in the brain, it is suggested to play a role in synaptic pool maintenance and vesicle trafficking [6,7].

Interestingly, in addition to α-syn, several heat shock proteins are co-localized in amyloid plaques, including α-crystallin. Studies have shown an upregulation in the brain of αB-crystallin in PD, Alzheimer´s disease and Creutzfeldt-Jacob disease [8,9]. αB-Crystallin has been shown to be a potent *in vitro* fibril formation inhibitor of α-syn wild-type, and mutant forms A30P and A53T [10]. Furthermore, αB-crystallin significantly reduces the *in vitro* aggregation of α-syn extracted and purified from the brain tissue of transgenic mice [11].

α-, β- and γ-Crystallins are the crucial structural proteins within the eye lens and are all responsible for its stability and transparency [12–15]. However, α-crystallins are the main protein components of mammalian eye lens, sharing sequence similarities with small heat-shock peptides. α-Crystallins are composed of two closely related 20 kDa polypeptide chains, αA (acidic) and αB (basic), that self-associate forming larger oligomeric complexes of 30–50 subunits [12–14]. The ratio of αA to αB is estimated to be 3:1 [12]. αA- and αB-crystallin are 173 and 175 amino-acid long peptides, respectively [16]. For a long time α-crystallin was thought to be exclusively expressed in the eye lens. Such idea was abandoned as it was discovered to occur in other tissues, such as, heart, skeletal tissue, kidney and brain [12,14,17]. α-Crystallins have an isoelectric point of 4 and have been suggested to start to lose subunits at pH 5–6 [18]. In the case of oligomeric β- and monomeric γ-crystallin, each is built up out of four Greek key motifs organized into two domains [15]. β-crystallins are highly polydisperse oligomers comprising 20–30 kDa subunits, forming complexes of 50–200 kDa [19] with an isoelectric point of ca. 8 [18]. In this work, we use the high molecular weight $\beta_H$-crystallin with an average molecular weight of 200 kDa. γ-Crystallins are found as monomers of ca. 20 kDa and with an isoelectric point between 7–8 [15,20]. Here, we use $\gamma_B$-crystallin, which has a molecular weight of ca. 21 kDa. Structurally the domains of both β- and γ-crystallin are similar [21].

In the present study, our goal was to investigate the influence of bovine α-, $\beta_H$- and $\gamma_B$- crystallin proteins on α-syn aggregation. This is possible as we have identified conditions governing reproducible α-syn aggregation kinetics [22]. The experiments were performed at mildly acidic pH, physiologically relevant as they mimic conditions found in particular cellular lumens, such as, endosomes and lysosomes. Under such experimental conditions, secondary nucleation processes are enhanced and strongly accelerate the aggregation kinetics of α-syn and as a result are the main source of new aggregates [4,22]. As a result of a high rate constant for secondary nucleation, nuclei generated from secondary nucleation usually dominate over primary nucleation from very early in the aggregation reaction an onwards [23,24]. For comparison purposes, at neutral pH, the rate constant of elongation and higher-order assembly of fibrils was found to be much higher than primary nucleation and secondary processes [4].

## Results

### The effect of α-crystallin proteins on α-syn fibril formation

To investigate the effect of α-crystallin proteins on α-syn fibril formation, aggregation kinetics was monitored using thioflavin-T (ThT) as a probe to detect amyloid formation. The aggregation experiments were performed at mildly acidic pH (pH 5.5) in non-binding PEGylated plates under quiescent conditions. Under these experimental conditions, primary nucleation of α-syn is undetectably slow. For that reason, preformed fibrils often termed seeds, are added to the sample reaction in known amounts (counted as monomer equivalents), to trigger aggregation by enhancing secondary processes [22]. For this, a fixed α-syn monomer concentration in the presence of three different α-syn seed concentrations was incubated with α-crystallin with concentrations ranging from 0 to 2 mg/ml (Fig 1A–1C). In the absence of α-crystallin

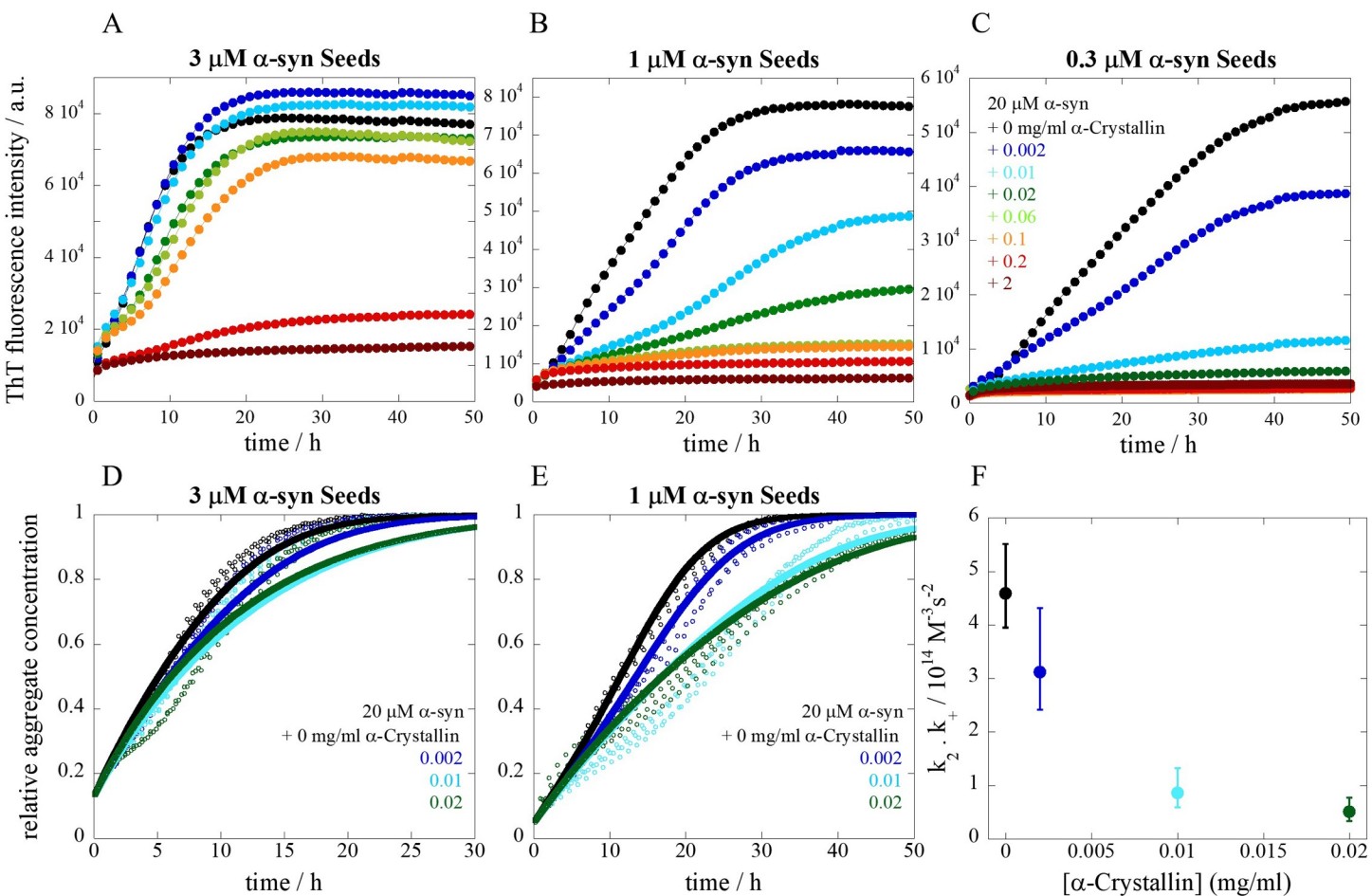

**Fig 1. Seeded aggregation kinetics of α-syn in the presence of α-crystallin.** The aggregation kinetics of 20 μM α-syn monomer was monitored using ThT fluorescence in the presence of three different seed concentrations **(A)** 3 μM, **(B)** 1 μM and **(C)** 0.3 μM with α-crystallin concentrations ranging from 0.002 to 2 mg/ml in 10 mM MES buffer pH 5.5 under quiescent conditions at 37°C with color codes shown in panel C. The figures show the average trace of at least three experimental repeats. The following panels show normalized aggregation kinetic traces for 20 μM α-syn monomer in the presence of **(D)** 3 μM and **(E)** 1 μM seed with α-crystallin concentrations ranging from 0–0.02 mg/ml. The figures show the experimental repeats dotted with the fits as solid lines. **(F)** Fitted values of the rate constant product $k_2k_+$ as a function of α-crystallin concentrations.

(black traces, Fig 1), the aggregation takes place within a few hours, in agreement with earlier findings [4,22]. By definition, secondary nucleation leads to an acceleration of the aggregation at early time points, which is manifested through observable ThT sigmoidal traces that have concave shapes [4,22]. Addition of α-crystallin progressively retards and even inhibits fibril formation within the time frame of the experiment. The effect is most evident at the lowest α-syn seed concentration (Fig 1C). In the samples where aggregation occurs in the presence of α-crystallin, the ThT plateaus exhibit lower fluorescence intensities which suggests a decrease in the levels of ThT-positive α-syn aggregates. This may be a result of the inhibitory effect or an interference with the ThT signal.

To ensure that the observed decrease in ThT fluorescence intensity with increasing amounts of α-crystallin (Fig 1) is due to a decrease in total mass of α-syn aggregates formed, the amount of monomeric α-syn was measured along the aggregation reaction (S1 Methods in S1 Text). This experiment consisted of monitoring the aggregation kinetics using ThT and in parallel following monomer depletion for samples taken at different time points by measuring UV absorbance at 280 nm both in the presence and absence of α-crystallin (S1 Fig). In the

absence of α-crystallin, for both seed concentrations tested (0.3 and 3 μM) we observe that when fibrils are formed, the amount of monomer concentration in solution decreases. This observed decrease in monomer concentration in solution measured at different time points coincides with the strong increase in ThT fluorescence intensity. In the presence of low α-crystallin (0.01 mg/ml) and low amount of seeds, there is no observable decrease in α-syn monomer concentration over 40 h, which is consistent with no increase in ThT fluorescence in the same time frame. In the case of high α-crystallin (0.05 mg/ml) and high α-syn seed concentrations, ThT fluorescence only shows a slight increase in intensity at the beginning of the aggregation reaction, matching the α-syn monomer concentration which also slightly decreases. It is evident from this experiment that the presence of α-crystallin leads to a decrease in the total amount of aggregates formed.

## Kinetic analysis

The experimental data for the two highest seed concentrations tested (3 and 1 μM) in the range where aggregation occurred, up to 0.02 mg/ml α-crystallin, was analyzed by fitting an aggregation kinetic model that includes secondary nucleation and elongation (Fig 1D and 1E) [25]. The aim was to gain from the experimental data information related to the effect of α-crystallin on the rate constants for nucleation and elongation. The successful determination of such mechanistic insights rely heavily on the purity of the protein samples and on the reporter dye (ThT) in the sense that there is a linear correlation between signal and total aggregate mass. As the reaction occurs in the presence of seeds, the contribution from primary nucleation ($k_n$) is overruled and therefore its rate constant becomes negligible [4,22]. The resulting fits provide estimates of the product of the rate constants of elongation ($k_+$) and secondary nucleation ($k_2$). The obtained values of the product $k_2k_+$ are plotted as a function of α-crystallin concentration in Fig 1F. We find that $k_2k_+$ decreases progressively with increasing α-crystallin concentration.

To facilitate the understanding on a molecular level, it is useful to provide the crystallin concentration not only in mg/mL but also in molar units. Here it is however worth mentioning that at this pH it has been suggested that the native α-crystallin may start to lose subunits [18], and therefore a mixture of (mainly) oligomeric and (a smaller amount of) monomeric entities with unknown exact ratio might be present. A 2 mg/ml solution of α-crystallin would thus approximately correspond to 2.5 μM of native α-crystallin (i.e. in its oligomeric form) or 100 μM monomeric mixture of αA/αB-crystallins (assuming that an oligomer consists on average of about 40 subunits) in the extreme but unrealistic case of full dissociation of native α-crystallin, respectively.

## The effect of $\beta_H$- and $\gamma_B$-crystallins on α-syn aggregation

Similar kinetic experiments were also performed with $\beta_H$- and $\gamma_B$-crystallin only for the intermediate α-syn seed concentration (1 μM). For oligomeric $\beta_H$-crystallin the inhibitory effect was shown to be similar to that of α-crystallin (Figs 1B vs 2A). For monomeric $\gamma_B$-crystallins the inhibition of fibril formation was slightly less efficient (Fig 2B).

## Trap and seed kinetic experiment

A set of experiments referred to as the trap and seed were performed to further evaluate if the effect of crystallins is indeed linked to the catalytic nature of α-syn fibrils (Fig 3). α-Syn seeds pre-incubated in the presence and absence of α- or $\gamma_B$-crystallin were trapped by filtration in low-binding GH Polypro (GHP) membrane filter plates with a 200 nm cutoff. Freshly purified α-syn monomer was then added and incubated with the trapped fibrils for 2 h and again filtered. The flow-through was collected, supplemented with ThT and placed in a plate reader at 37˚C under quiescent conditions (Fig 3). It is observed that α-syn seeds that were not pre-incubated with crystallins

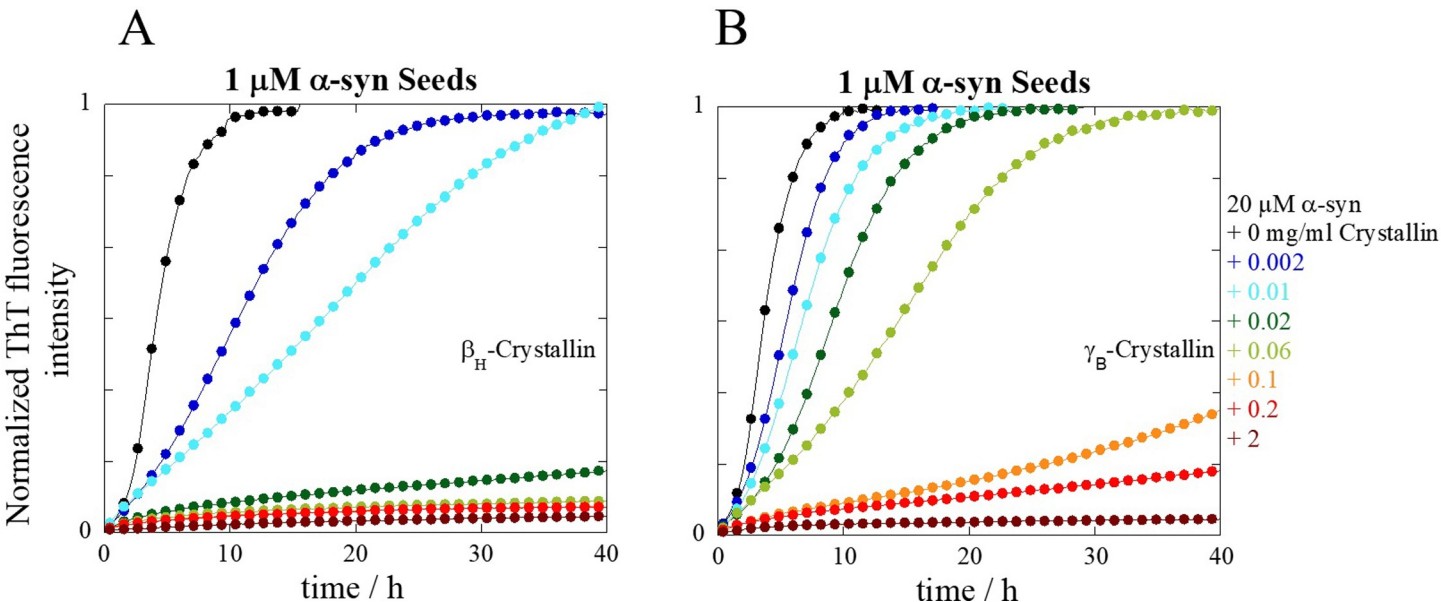

**Fig 2. Seeded aggregation kinetics of α-syn monomer in the presence of β_H- and γ_B-crystallin.** The aggregation kinetics of 20 μM α-syn was monitored using ThT for α-syn alone (black traces) and in the presence of **(A)** β_H- or **(B)** γ_B-crystallin at concentrations ranging from 0.002 to 2 mg/ml for one α-syn seed concentration, 1 μM, in 10 mM MES buffer pH 5.5 under quiescent conditions at 37°C. The experimental average trace of at least two repeats is shown. For stoichiometric purposes, 2 mg/ml of the high molecular weight oligomeric β_H-crystallin approximately corresponds to a molar concentration of 10 μM, while for monomeric γ_B-crystallin it is roughly 95 μM.

trigger monomer aggregation (Filtrate 1; Fig 3A). This result is consistent with the interpretation that the trapped fibrils provide an autocatalytic surface for the monomer added, generating small oligomeric species that pass through the filter and further grow and catalyze aggregation of the remaining monomer in solution. This is also consistent with previous studies [22]. Relevant to highlight is that the GHP filter plates used in this experiment do not trigger α-syn aggregation, shown by incubating monomeric α-syn in the absence of seeds in these plates (Control). For both cases, seeds pre-incubated with 0.2 and 2 mg/ml of either α- or γ_B-crystallin were no longer capable of seeding the aggregation reaction during the time frame of the experiment (Filtrates 2 and 3 for α-crystallin; Filtrates 4 and 5 for γ_B-crystallin, respectively; Fig 3B and 3C).

### The effect of α-crystallin depends on when it is introduced to the aggregation reaction

In addition, we investigated the ability of α-crystallin to inhibit α-syn fibril formation when added at different time points along the aggregation reaction (Figs 4 and S2). Additions made at early time points have a strong effect on α-syn aggregation, retarding aggregation when compared to α-syn alone (black traces). The addition of α-crystallin seems to retard/inhibit aggregation if added before $t_{1/2}$, which is the time at which the ThT fluorescence reaches 50% of the total fluorescence intensity. This effect is seen for both seed concentrations. These results further strengthen that the effect of crystallin proteins may be on the autocatalytic nature of α-syn seeds.

### Discussion

There are many mechanisms suggested to be involved in the development of PD, such as, accumulation of misfolded protein aggregates, incompetent protein clearance, mitochondrial dysfunction, oxidative stress, neuroinflammation and genetic mutations [5]. With aging

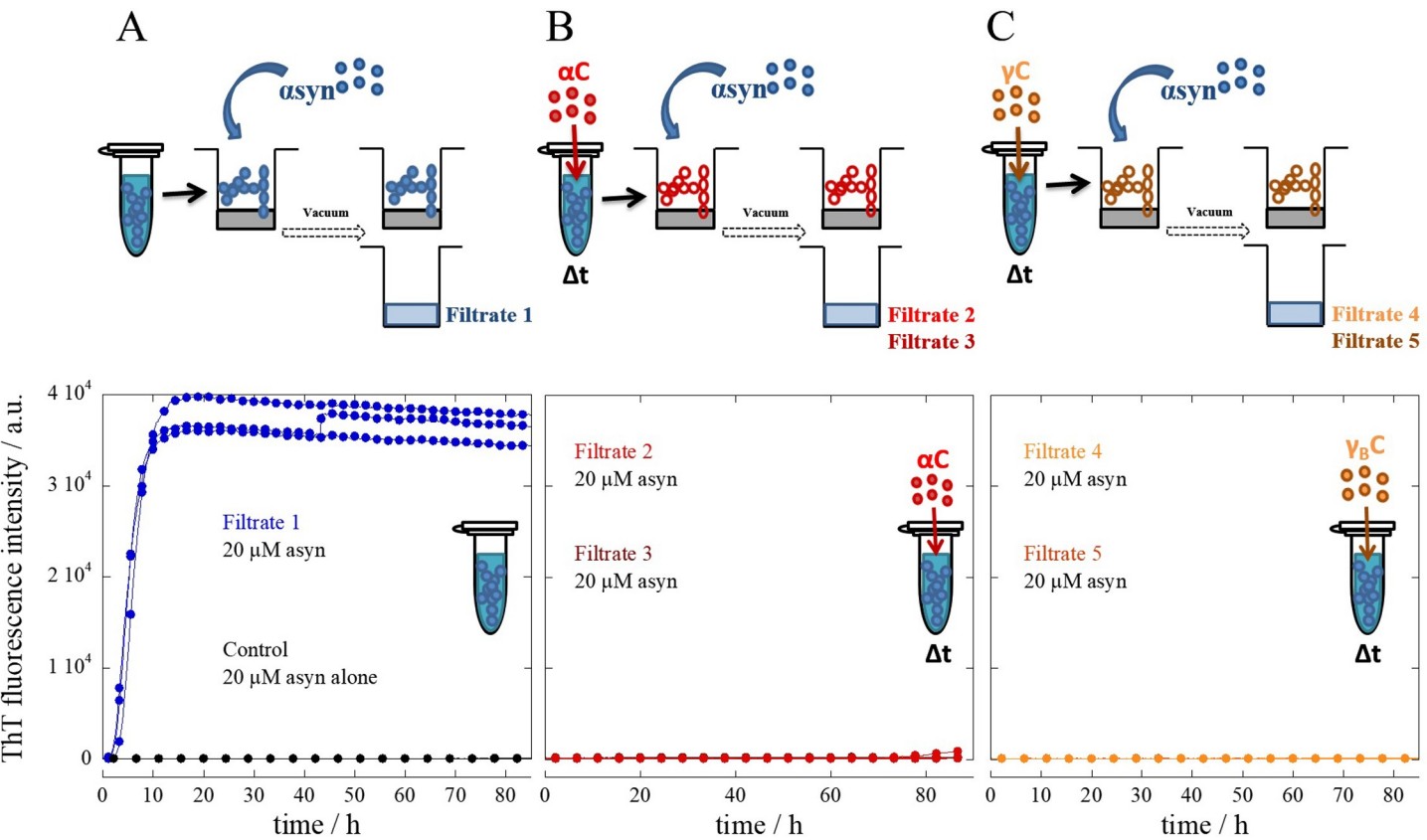

**Fig 3. Trap and seed kinetic experiment.** 3 μM α-syn seeds without and with pre-incubation with α- or γ_B-crystallin were trapped by filtration in GHP filter membrane plates with a 200 nm cutoff. The trapped fibrils without any pre-treatment were then incubated for 2 h with freshly purified α-syn monomer and newly filtered ((**A**) Filtrate 1). The same procedure was performed for the trapped fibrils that had been pre-incubated for 2 h with two concentrations 0.2 and 2 mg/ml of either α-crystallin ((**B**) Filtrate 2 and 3, respectively) or γ_B-crystallin ((**C**) Filtrate 4 and 5, respectively). The flow-through was collected in non-binding PEGylated plates supplemented with ThT and monitored in a plate reader under quiescent conditions at 37˚C. The figures show the individual aggregation kinetic traces of at least three experimental repeats.

chaperone activity becomes disrupted affecting both the appropriate levels and conformations of proteins [26]. Chaperone activity has been vastly investigated in association to amyloid-related disorders and shown in several studies capable of inhibiting fibril formation for different amyloid peptides, such as, α-syn, Aβ and PolyQ peptides by affecting different microscopic events of the aggregation mechanism [27–31]. Interestingly, αB-crystallin which displays chaperone-like activity accumulates in the central nervous system and co-localizes in Lewy bodies of patients with PD [10,32] and was found to be a major component of ubiquitinated inclusions bodies in human degenerative diseases [33]. Both αB-crystallin and hsp27 are expressed at elevated levels in the brains of patients with Alzheimer´s disease [34–36].

We have here investigated the effect of crystallins on α-syn aggregation. Aggregation kinetic experiments were conducted at mildly acidic pH, physiologically relevant as it mimics solution conditions found in intracellular compartments, such as, lysosomes and endosomes. The aggregation mechanism of α-syn at mildly acidic pH is dominated by monomer-dependent secondary nucleation [4,22]. This leads to the generation of larger amounts of smaller oligomeric species which are thought to be the most cytotoxic species. Curiously, it has been shown that α-crystallins prevent both β- and γ-crystallins from heat-induced aggregation [13,37], while all three bovine crystallins form β-sheet enriched amyloid fibrils under denaturing and acidic pH conditions [19]. For that reason, a preliminary experiment was conducted incubating bovine crystallins at mildly

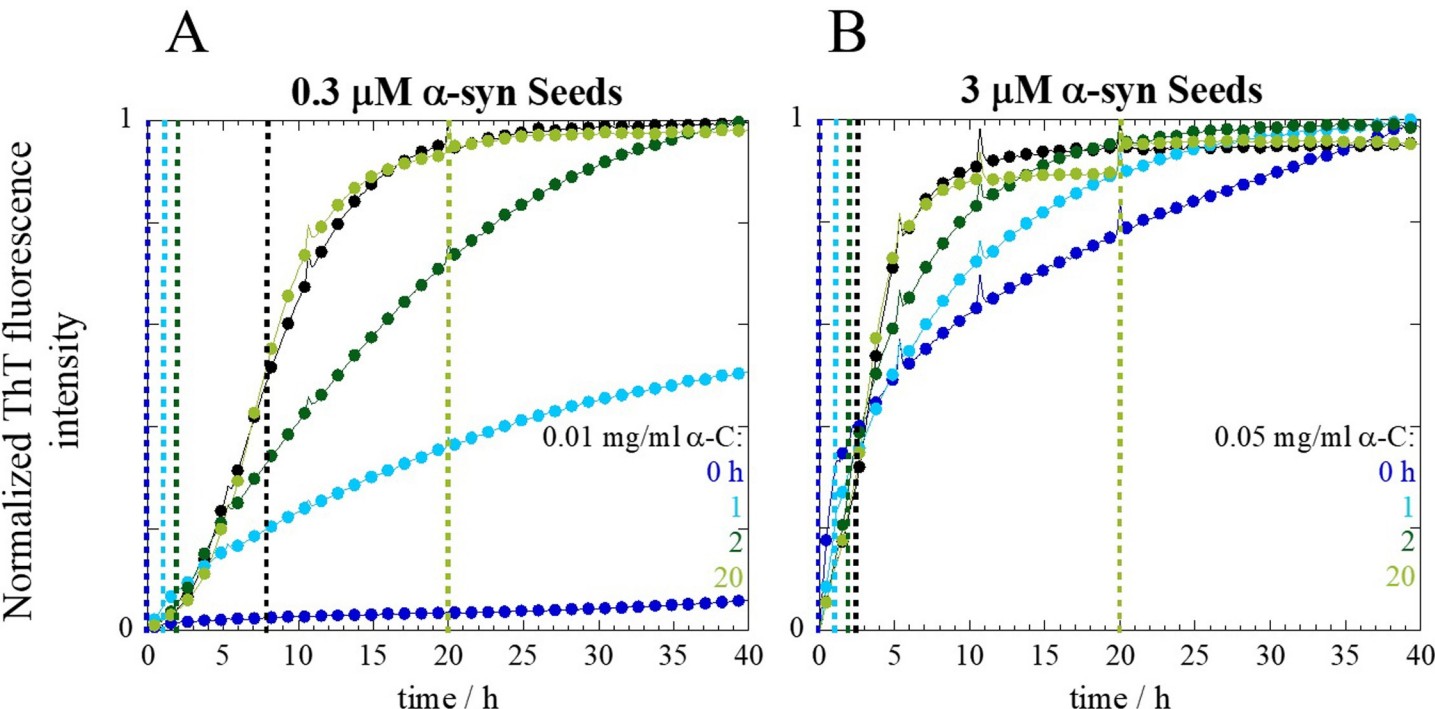

**Fig 4. Aggregation kinetics of α-syn in the presence of α-crystallin added at different time points along the aggregation reaction.** Seeded aggregation kinetics was monitored using ThT for 20 μM α-syn alone (black traces) and in the presence of two different α-crystallin concentrations **(A)** 0.01 mg/ml and **(B)** 0.05 mg/ml in 10 mM MES buffer pH 5.5 under quiescent conditions at 37˚C. α-Crystallin was added at different time points along the aggregation reaction, 0, 1, 2 and 20 h, indicated by vertical dashed lines with matching color code. The $t_{1/2}$ is also indicated by a black dashed vertical line. The average of at least three experimental repeats is shown.

acidic conditions (pH 5.5) supplemented with ThT and placed in a plate reader to monitor for fibril formation for ~45 h. No increase in ThT fluorescence intensity was observed (data not shown). All bovine crystallins here investigated were shown to inhibit α-syn aggregation evident from the ThT kinetic experiments. While the effect of all three crystallins were quite similar, oligomeric α- and $β_{H-}$ were shown to be slightly more efficient (Table 1). The observed extensive inhibitory effect of crystallin proteins far below equimolar amounts of α-syn is a first indication that their mechanism of action is not to interact with and reduce free α-syn monomer in solution but possibly to interact with the aggregated species affecting there catalytic nature. It is relevant to mention that at these experimental solution conditions, α-crystallins have been reported to start to lose subunits [18] and thus might exist as a mixture of oligomers in equilibrium with a small

**Table 1. Lowest crystallin protein concentrations which efficiently inhibit α-syn fibril formation in the presence of 1 μM α-syn seeds (Figs 1B and 2A and 2B).**

| | Lowest concentration which efficiently inhibits α-syn fibril formation in the presence of 1 μM α-syn seeds | | |
|---|---|---|---|
| Crystallin | Mass concentration (mg/ml) | Molar concentration of oligomers (μM) | Molar concentration of monomers/subunits (μM) |
| α[a] | 0.06 | 0.075 | 3 [b] |
| $β_H$[c] | 0.02 | 0.1 | 0.8 [d] |
| $γ_B$[e] | 0.1 | --- | 4.75 |

(a) Occurring mainly in its oligomeric form (native α-crystallin). Mw ca. 800 kDa.

(b) On average ca. 40 subunits/oligomer

(c) Occurring in its oligomeric form. Mw ca. 200 kDa

(d) On average ca. 8 subunits/oligomer

(e) Occurring as monomers. Mw ca. 21 kDa

amount of their subunits (monomers). The chaperone activity of α-crystallin has been previously attributed to its dissociation into smaller species which expose higher degree of hydrophobicity compared to the larger oligomers [26]. It is generally accepted that α-crystallin suppresses aggregation through hydrophobic interactions [16]. Interestingly, here it is shown that bovine crystallin proteins influence the aggregation kinetics either as monomeric (α-subunits and/or $\gamma_B$) or oligomeric (α and/or $\beta_H$) entities, as well as, with a positive ($\beta_H$ and $\gamma_B$) or negative (α) net charge. In addition, similar ThT experiments were performed with a control protein, lysozyme, resulting in a similar (though slightly reduced) potency of inhibition of fibril formation when compared to crystallins (S3 Fig). These results suggest that this inhibitory mechanism displayed by crystallin proteins appear to be more generalized and dependent on generic physical-chemical parameters rather than a specialized chaperone function.

The chaperone-like mechanism of crystallins remains unclear and many different scenarios have been proposed. αB-crystallin has been suggested to interact transiently with monomeric species of ApoC-II, inducing fibril-incompetent conformations promoting fibril dissociation [38]. αB-crystallin has also been shown to bind to amyloid fibrils of Aβ, α-syn and insulin and inhibit elongation [39,40]. Moreover, the inhibition of α-crystallin has been postulated to be a result of an interaction with nucleus species preventing its progression into amyloid fibrils [8]. To elucidate the inhibitory mechanism of crystallins a trap and seed kinetic experiment was performed. α-Syn seeds without and with a pre-incubation with crystallin proteins were trapped in filter membrane plates and incubated for a 2 h period with freshly purified α-syn monomer. After this incubation period, the samples were newly filtered, supplemented with ThT and aggregation kinetics monitored. In the absence of crystallin, α-syn seeds were shown to be catalytic, suggesting that in a reaction between α-syn seeds and monomer, smaller oligomeric species are generated. These aggregated species were small enough to pass through the filter and further grow and aggregate the remaining monomer. However, in the cases where α-syn seeds had been pre-incubated with either α- or $\gamma_B$-crystallin, these seeds no longer displayed this catalytic effect. This result implies that the inhibitory effect may arise as a result of an interaction between crystallins and mature α-syn fibrils. One can hypothesize that crystallins may interact along the surfaces and free ends of the fibrils inhibiting both elongation and monomer-dependent secondary nucleation. These findings are highly relevant as secondary nucleation increases rapidly both the load of oligomeric species and fibrils. This also falls in line with previous findings where the chaperone-activity of α-crystallins was suggested to act as "holdases", holding proteins in large soluble aggregates [26]. Inhibition of secondary nucleation of α-syn was shown to occur by the homologous protein β-syn [41]. Several studies have also shown that substoichiometric concentrations of transthyretin (TTR) [42], Brichos [30], antibody fragments [43] and several small molecules [44] inhibit secondary nucleation of Aβ peptide.

Finally, we then investigated the ability of α-crystallin to inhibit α-syn fibril formation when added at different time points along the aggregation reaction. Early additions of α-crystallin, before $t_{1/2}$, were shown capable of retarding aggregation. However, additions of α-crystallin after $t_{1/2}$ no longer displayed such effect. Similar observations have been reported for ApoC-II [8]. Furthermore, inhibition studies of amyloid-β fibril formation using either polymeric nanoparticles or Brichos domains also found $t_{1/2}$ to be the critical point, where additions prior to $t_{1/2}$ affected fibril formation opposing to additions made at time points after $t_{1/2}$ [45,46].

## Material and methods

### α-syn expression and purification

Human α-syn was expressed in E. coli and purified using heat treatment, ion exchange and gel-filtration chromatography, as previously described [47]. A key procedure to achieving

reproducible experiments is to use pure samples as starting material. Gel-filtration is a crucial step to isolate pure monomeric α-syn in degassed 10 mM MES buffer pH 5.5. Only protein sample corresponding to the central region of the peak was collected. Freshly purified α-syn was prepared and handled always on ice to avoid initiation of the aggregation process. The peptide concentration was determined by absorbance at 280 nm using an extinction coefficient $\varepsilon = 5800$ $l \cdot mol^{-1} cm^{-1}$.

## Crystallin protein preparation

Crystallins (α, $\beta_H$ and $\gamma_B$) were obtained using the procedure established by Thurston [48; S2 Methods in S1 Text] from fresh calf-lenses provided as a by-product from a slaughterhouse. The eye-lenses were stored at 4°C in a 52.4 mM Phosphate buffer consisting of 34 mM $Na_2HPO_4$; 18.4 mM $NaH_2PO_4$; 1 mM EDTA (prevents enzymatic degradation)); 1 mM DTT (prevents oxidation) and 0.02 wt% $NaN_3$ (prevents bacterial growth). This buffer, that is also used as storage medium for the extracted α- and $\beta_H$-crystallins, has a pH of 7.1 and an ionic strength of c.a. 175 mM. The $\gamma_B$–rich lens nuclei are separated from the α- and β-rich coronas (cortical) and transferred to a NaAc-buffer consisting of 275 mM acetic acid, 100 mM NaOH and 0.02 wt% $NaN_3$ with a pH of 4.5. The lens cell walls are broken down using an electrical grinder and the final slurry filtered to retrieve a mixture of eye-lens proteins. The individual crystallins, in the case of α and $\beta_H$, and a mix of different γ were extracted by passing the different mixtures through a size-exclusion chromatography (SEC) column containing Superdex 200 prep grade and eluting with excess buffer. The different fractions corresponding to α- and $\beta_H$-crystallins were collected and stored at 4°C (S4 Fig). The γ-mixture was filtered and degassed before passed through a SP Sepharose Fast Flow ion-exchange (IEX) column and the proteins (including $\gamma_B$) individually eluted using a NaAc-buffer with a salt gradient at pH 4.8 consisting of 275 mM Acetic acid; 167.5 mM NaOH; 0.02wt% $NaN_3$; 0–325 mM NaCl (S5 Fig). The protein buffers were exchanged using Amicon Ultra Centrifugal Devices, 10 kDa for α and β and 3 kDa for $\gamma_B$, to 10 mM MES buffer pH 5.5. The protein concentrations were determined by absorbance at 280 nm, using extinction coefficients α = 0.845 [49], $\beta_H$ = 2.3 [50] and $\gamma_B$ = 2.18 [48]. The hydrodynamic radius ($R_h$) of α- and $\beta_H$-crystallin proteins in dilute solutions obtained from Dynamic Light Scattering (DLS) experiments at pH 7.1 (phosphate buffer used during the SEC preparation procedure) and pH 5.5 (MES buffer used for the aggregation kinetics experiments), were shown to remain almost constant (S6 Fig and S7 Fig, respectively; S3 Methods in S1 Text).

## Thioflavin-T aggregation kinetics assay

To monitor fibril formation, 100 μl samples were aliquoted in 96-well non-binding PEGylated plates (Half-area, 3881 Corning plates), supplemented with 20 μM ThT and sealed with a plastic film to avoid evaporation. Plates were incubated at 37°C under quiescent conditions in a plate reader (FluoStar Omega, BMG Labtech, Offenburg, Germany).

The experimental data were fitted by the following equation using the Amylofit online interface [25]:

$$\frac{M(t)}{M(\infty)} = 1 - \left( \frac{B_+ + C_+}{B_+ + C_+ e^{\kappa t}} \frac{B_- + C_+ e^{\kappa t}}{B_- + C_+} \right)^{\frac{k_\infty^2}{\kappa K_\infty}} e^{-k_\infty t}$$

Where,

$B_\pm = (k_\infty \pm \check{k}_\infty) / (2\kappa)$

$C_\pm = \pm \lambda^2 / (2\kappa^2)$

$\kappa = \sqrt{\{2k_+ k_2 m(0)^{n2+1}\}}$

$$\lambda = \sqrt{\{2k_+ k_n m(0)^{n_c}\}}$$
$$k_\infty = \sqrt{\{2\kappa^2/[n_2(n_2+1)]+2\lambda^2/n_c\}}$$
$$\check{k}_\infty = \sqrt{\{k_\infty^2 - 4C_+ C_- \kappa^2\}}$$

## Trap and seed assay

α-Syn seeds were made from 40 μM monomeric α-syn under shaking conditions at 37°C in Eppendorf tubes. Versatile GH-Polypro filter membranes of low-binding AcroPrep 96-well filter plates (Pall Life Sciences, Ann Arbor, MI) were initially washed with experimental buffer and saturated with monomeric α-syn. After, α-syn seeds with and without crystallin pre-treatment (2 h incubation) were trapped on these filter plates by filtration applying vacuum for 10 s and discarding the flow-through. The filtration was done using a MultiScreenHTS vacuum (Millipore) manifold. Monomeric α-syn was incubated with the trapped aggregates and newly filtered. This flow through was collected in 96-well non-binding PEGylated plates, supplemented with ThT and fluorescence intensity monitored in a plate reader at 37°C under quiescent conditions.

## Supporting information

**S1 Fig. α-syn monomer depletion during the aggregation reaction.** The seeded aggregation kinetics for 20 μM α-syn alone (black lines) and in the presence of α-crystallin (blue lines) was monitored using ThT. In parallel, samples along the aggregation reaction were aliquoted and centrifuged for 10 minutes at 13,000 g. The monomer concentration in the supernatant was measured by UV absorbance at 280 nm (open symbols and dashed lines). **(A)** 0.3 μM α-syn seeds alone and in the presence of 0.01 mg/ml α-crystallin; **(B)** 3 μM α-syn seeds alone and in the presence of 0.05 mg/ml α-crystallin. The experimental averages of at least three experimental repeats are shown. Important to highlight, the small artifacts seen in the ThT traces are due to the opening of the plate reader to remove samples for monomer concentration determination.
(TIF)

**S2 Fig. Aggregation kinetics of α-syn in the presence of α-crystallin added at different time points along the aggregation reaction.** Seeded aggregation kinetics was monitored using ThT for 20 μM α-syn alone (black traces) and in the presence of two different α-crystallin concentrations **(A)** 0.01 mg/ml and **(B)** 0.05 mg/ml in 10 mM MES buffer pH 5.5 under quiescent conditions at 37°C. α-Crystallin was added at different time points along the aggregation reaction, 0, 1, 2, 5, 10 and 20 h. The average of at least three experimental repeats is shown.
(TIF)

**S3 Fig. Seeded aggregation kinetics of α-syn in the presence of lysozyme.** The aggregation kinetics of 20 μM α-syn monomer was monitored using ThT fluorescence in the presence of 1 μM seed concentration with Lysozyme concentrations ranging from 0.002 to 2 mg/ml in 10 mM MES buffer pH 5.5 under quiescent conditions at 37°C with color codes shown to the left. The figures show the average trace of at least three experimental repeats.
(TIF)

**S4 Fig. Size-exclusion chromatography of the cortical extract to retrieve α- and β$_H$-crystallin.** Chromatogram of cortical calf eye lens extract after passing through the SEC column which allows for the extraction of α- and β$_H$-crystallin, with the collected fractions indicated by the vertical lines. Also visible are low molecular weight β variants, as well as a mixture of γ-

crystallin variants.
(TIF)

**S5 Fig. Size-exclusion and ion exchange chromatography of the nuclear extract to retrieve γ$_B$-crystallin. (a)** A typical chromatogram of the nuclear calf eye lens extract after having passed a SEC column where a mixture of γ-crystallin variants is obtained by collecting the indicated elution volume. **(b)** Resulting chromatogram from an IEX column after passing the mixture of γ-crystallin variants shown in (a). This allows for the collection of purified γ$_B$-crystallin while the other variants of γ-crystallin are discarded.
(TIF)

**S6 Fig. Hydrodynamic radius of α-crystallin proteins at pH = 7.1 and 5.5.** Correlation functions from DLS measurements on α-crystallin solutions at pH 7.1 (left) and pH 5.5 (right). Open symbols: data; red line: $2^{nd}$ order cumulant fit. See text for details (S3 Methods in S1 Text).
(PDF)

**S7 Fig. Hydrodynamic radius of β$_H$-crystallin proteins at pH = 7.1 and 5.5.** Correlation functions from DLS measurements on β$_H$-crystallin solutions at pH 7.1 (left), and pH 5.5 (right). Open symbols: data; red line: $2^{nd}$ order cumulant fit. See text for details.
(PDF)

**S1 Text.**
(DOCX)

## Acknowledgments

We would like to acknowledge Emma Sparr and Sara Linse for expert discussion, inputs in the experiments, data analysis and paper revision. We are particularly grateful to Alessandro Gulotta for the precious help with the Dynamic Light Scattering experiments.

## Author Contributions

**Conceptualization:** Ricardo Gaspar.

**Formal analysis:** Ricardo Gaspar, Anna Stradner.

**Investigation:** Ricardo Gaspar, Tommy Garting.

**Methodology:** Tommy Garting.

**Supervision:** Anna Stradner.

**Writing – original draft:** Ricardo Gaspar.

**Writing – review & editing:** Tommy Garting, Anna Stradner.

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
