## [Decision Letter · Decision Letter 0]

15 May 2020

PONE-D-20-11144

Eye lens crystallin proteins inhibit the autocatalytic amyloid amplification nature of mature α-synuclein fibrils

PLOS ONE

Dear Dr Ricardo Gaspar,

Thank you for submitting your manuscript to PLOS ONE. As you can see below, the reviewers overall like the study, but their comments were not completely favorable throughout.  They made some suggestions for changes which I would.like to invite you to consider.  I'm sure you are aware that PLOS ONE has slightly different publication criteria than most journals, and I feel this issue may impact your responses.  I am looking forward to a revised version of the manuscript that responds to the points raised during the review process.  

We would appreciate receiving your revised manuscript by Jun 29 2020 11:59PM. To enhance the reproducibility of your results, we recommend that if applicable you deposit your laboratory protocols in protocols.io, where a protocol can be assigned its own identifier (DOI) such that it can be cited independently in the future. For instructions see: http://journals.plos.org/plosone/s/submission-guidelines#loc-laboratory-protocols

We look forward to receiving your revised manuscript.

Kind regards,

Peter Schuck

Academic Editor

PLOS ONE

Journal Requirements:

Reviewers' comments:

Reviewer's Responses to Questions

**Comments to the Author**

1. Is the manuscript technically sound, and do the data support the conclusions?

Reviewer #1: Partly

Reviewer #2: No

Reviewer #3: Yes

2. Has the statistical analysis been performed appropriately and rigorously? 

Reviewer #1: Yes

Reviewer #2: N/A

Reviewer #3: Yes

3. Have the authors made all data underlying the findings in their manuscript fully available?

Reviewer #1: Yes

Reviewer #2: Yes

Reviewer #3: No

4. Is the manuscript presented in an intelligible fashion and written in standard English?

Reviewer #1: Yes

Reviewer #2: Yes

Reviewer #3: Yes

5. Review Comments to the Author

Reviewer #1: This paper describes the effects of purified bovine lens crystallin fractions on a-synuclein fibril formation in vitro. All three fractions tested inhibited fibril formation in similar ways. The experimental design is straightforward.

The effect of aB-crystallin in this type of experiment has been shown before. aB is a bona fide small heat shock protein and has wide activities in preventing aggregation of unfolding proteins. However, b- and g-crystallins are quite different proteins with no structural relationship to aB/sHSPs. It is interesting that they have such similar effects. It would suggest that the inhibition is not directly related to the sHSP chaperone properties of aB.

It has been suggested (https://www.ncbi.nlm.nih.gov/pubmed/24582830) that all crystallins, even of different evolutionary origins, have properties, probably related to the organization of the protein surface, that allow them to exist under conditions of molecular crowding without aggregation. These properties may extend to stabilizing interactions with other cellular proteins, such as those of the cytoskeleton. However, in these experiments, it is not clear whether this is a specialized function of crystallins or a generalized effect of heterologous proteins interfering with fibril assembly. This would be clearer if the authors included a negative control using other soluble globular proteins (say lysozyme) to show they have no effect in the assay.

Reviewer #2: In the manuscript, the authors demonstrated that alpha-, beta-, and gamma-crystalin inhibit amplification of aS amyloid in vitro. The conclusion is not entirely surprising, as alpha-crystalin was previously shown to act as molecular chaperon against amyloid formation. However, the reviewer does not support its publication in the journal, since the current manuscript contains multiple flaws, and represents only phenomenological descriptions without insight into the nature of inhibition. 

1.Regarding to Figure 1, the reviewer recommends the authors to repeat the same experiment at pH 7.4, where other processes of amyloid formation are operative. Such experiment provides insights into the nature of amyloid inhibition by crystalins, and improves the physiological relevance of the experiment.

2. In the experiment shown in Figure S1, the authors examined the residual amount of aS monomer in the supernatant. This experiment is of importance, because several small compounds decrease ThT fluorescence without reducing the total amount of amyloid fibrils. However, two concerns arise regarding the experimental procedure. At first, it remains unclear whether the supernatant is truly devoid of aS amyloid fibrils (regarding to this point, centrifugation time and speed are not provided in the manuscript). I recommend the authors to measure ThT fluorescence before and after the centrifugation to examine what percentage of amyloid fibrils are precipitated by the centrifugation. Secondary, UV absorbance is not an accurate measure of protein concentration in a mixture containing two different proteins. Protein concentration should be determined by SDS-PAGE after separating aS from crystalins and measuring the band intensity.

3.The trap and seed kinetic experiment is tricky and less understandable. At first, it is difficult for me to understand which solutions marked by "filtrate 1", "control", "filtrate 2", etc., are filtrated once or twice. Furthermore, this experiment seems to contain a critical flaw: it was not proven that crystalins do not pass through the 200 nm filter. If not, crystalins will present in the trapped fraction at a high concentration, and inhibit generation of small oligomer in the next reaction. Crystalins form multimers, possibly larger than 200 nm. Furthermore, crystalins can interact with aS amyloid fibrils, which also potentially inhibit their flow-through.

4.In the abstract and discussion sections, the author hypothesized that crystalins might interact with aS fibrils. This hypothesize should be tested to provide insight into the nature of inhibition. For example, as described in doi: 10.1021/cn500019c, co-sedimentation assay is a simple method to detect interaction between amyloid fibrils and proteins. SPR analysis is alternative.

Reviewer #3: In this manuscript, Gaspar et al. investigate the effects of three types of crystallin proteins on the seeded amyloid fibril formation of alpha-synuclein at mildly acidic pH. The main finding is that the reaction of fibril formation is inhibited by these chaperone-like proteins. While it is not clear how relevant crystallins are for alpha-synuclein amyloid fibril formation in vivo, the study is nevertheless interesting and relevant. Not least because we still lack basic mechanistic understanding of alpha-synuclein amyloid formation under these conditions and any additional piece of information, also specific modes of action of inhibitors, is of value.

Overall I think the study is well-performed and the conclusions are mostly justified. However, I think that a few aspects need to be characterised/discussed in some more detail:

1) Some data needs to be shown from the purification of the proteins, such as SEC chromatograms, in order to give the reader a feel for how pure these preparations are, which are extracted from bovine eyes. Also, I would like to see a characterisation of the final chaperone preparations, at least in terms of their overall size distribution. Here, DLS, analytical ultracentrifugaion or microfluidic diffusional sizing would be appropriate techniques. Such measurements should be done both at pH 7, where the crystallins are extracted, as well as at pH 5.5, where the kinetic experiments are done.

2) Overall, the arguments that the crystallins inhibit secondary nucleation is supported by the data, in particular by the trap and seed experiments. However, the data also shows a different slope at time 0, which suggests an effect on elongation. This should be investigated in more detail, in order to be able to provide an estimate of the relative effects on elongation and secondary nucleation. Another noteworthy feature in the kinetic data is that the curves in the presence of inhibitor reach a plateau at lower ThT fluorescence intensities than the level in the absence of inhibtor. I guess for the amylofit analysis, the assumption was made that the ThT level is exactly proportional to the fibril concentration (this should be stated somewhere). That would mean that the inhibitor not only changes the kinetics, but also the final equilibrium position. This is interesting, as it should only happen with inhibitors that can interact with the monomer, and hence are able to shift the equilibrium. This seems not to be the case, however, for the crystallins. This should be commented on.

3) An estimate should be provided for the stoichimetry of fibril binding of the crystallins. For example one could incubate fibrils and crystallins and then spin them down and see how much of the crystallin is spund down together with the fibrils. This would be important to be able to compare its effect with that of other chaperones that can inhibit secondary nucleation in other amyloid systems, such as Abeta (e.g. Brichos chaperone). Also, there should be a somewhat more extensive discussion of the existing literature of secondary nucleation inhibitors in general and alpha-synuclein in particular, as some recent studies have presented efficient inhibition of alpha-synuclein secondary nucleation at highly sub-stoichiomtric ratio.

6. PLOS authors have the option to publish the peer review history of their article (what does this mean?). If published, this will include your full peer review and any attached files.

Reviewer #1: No

Reviewer #2: No

Reviewer #3: No

---

## [Author Response · Author response to Decision Letter 0]

5 Jun 2020

Resubmission of revised version of PONE-D-20-11144:

“Eye lens crystallin proteins inhibit the autocatalytic amyloid amplification nature of mature α-synuclein fibrils” by Ricardo Gaspar, Tommy Garting and Anna Stradner.

Dear Editor,

Thank you very much for the reviewer reports for our manuscript PONE-D-20-11144.

We are very happy with the positive feedback and grateful to the reviewers for the numerous thoughtful comments and suggestions to improve the manuscript.

We have now carefully and substantially revised the manuscript along these comments and feel that this has considerably increased the clarity and readability of the manuscript.

Please find below a point-by-point response to all the comments and criticism raised by the three reviewers. 

We hope that with these changes our manuscript is now acceptable for publication as a Research Article in PLOS ONE.

We thank you in advance for your time and your consideration of our submission.

Sincerely yours,

Ricardo Gaspar

Detailed response to the reviewers’ comments

The original comments by the reviewers are repeated in italic for clarity, our answers are highlighted in red, and the revisions made are highlighted in blue in this response letter and marked in yellow in the revised manuscript (pdf).

REVIEWERS’ COMMENTS: 

Reviewer #1: 

This paper describes the effects of purified bovine lens crystallin fractions on a-synuclein fibril formation in vitro. All three fractions tested inhibited fibril formation in similar ways. The experimental design is straightforward. The effect of aB-crystallin in this type of experiment has been shown before. aB is a bona fide small heat shock protein and has wide activities in preventing aggregation of unfolding proteins. However, b- and g-crystallins are quite different proteins with no structural relationship to aB/sHSPs. It is interesting that they have such similar effects. It would suggest that the inhibition is not directly related to the sHSP chaperone properties of aB. It has been suggested (https://www.ncbi.nlm.nih.gov/pubmed/24582830) that all crystallins, even of different evolutionary origins, have properties, probably related to the organization of the protein surface, that allow them to exist under conditions of molecular crowding without aggregation. These properties may extend to stabilizing interactions with other cellular proteins, such as those of the cytoskeleton. However, in these experiments, it is not clear whether this is a specialized function of crystallins or a generalized effect of heterologous proteins interfering with fibril assembly. This would be clearer if the authors included a negative control using other soluble globular proteins (say lysozyme) to show they have no effect in the assay.

Response: We thank the reviewer for the positive feedback and for pointing out this extremely interesting paper. We agree with the reviewer and have added an experiment with lysozyme (shown in supplementary information). We feel that with the data shown for all three crystallins and the negative control experiment from lysozyme, this strongly suggests that the inhibitory effect on fibril formation appears to be more generalized and dependent on generic physical-chemical parameters rather than a specialized chaperone function attributed to small heat-shock proteins, as alpha-crystallin. 

We have added a respective discussion to the revised manuscript. It now reads on page 12:

“In addition, similar ThT experiments were performed with a control protein, lysozyme, resulting in a similar (though slightly reduced) potency of inhibition of fibril formation when compared to crystallins (Fig S3 in SI). These results suggest that this inhibitory mechanism displayed by crystallin proteins appear to be more generalized and dependent on generic physical-chemical parameters rather than a specialized chaperone function.”

The following figure was added in supporting information (Fig S3):

“Fig S3. Seeded aggregation kinetics of α-syn in the presence of Lysozyme. The aggregation kinetics of 20 µM α-syn monomer was monitored using ThT fluorescence in the presence of 1 µM seed concentration with Lysozyme concentrations ranging from 0.002 to 2 mg/ml in 10 mM MES buffer pH 5.5 under quiescent conditions at 37ºC with color codes shown to the left. The figures show the average trace of at least three experimental repeats.”

Reviewer #2: 

In the manuscript, the authors demonstrated that alpha-, beta-, and gamma-crystallin inhibit amplification of aS amyloid in vitro. The conclusion is not entirely surprising, as alpha-crystallin was previously shown to act as molecular chaperon against amyloid formation. However, the reviewer does not support its publication in the journal, since the current manuscript contains multiple flaws, and represents only phenomenological descriptions without insight into the nature of inhibition. 

Response: We thank the reviewer for the feedback related to our manuscript. While the effect for alpha-crystallin may not be entirely surprising, this is the first study to the best of our knowledge that reports similar effects for beta- and gamma-crystallin. It is extremely interesting that crystallin proteins of different evolutionary origins and with no structural relationship share similar effects on fibril formation. This was very nicely postulated in a paper suggested by Reviewer #1 (https://www.ncbi.nlm.nih.gov/pubmed/24582830). The results here presented imply that the inhibitory effect on fibril formation may have origins in more generalized and generic physical-chemical parameters rather than specialized chaperone function attributed to small heat-shock proteins. This picture is also supported by negative control experiments using lysozyme, as suggested by Reviewer #1, and as detailed and discussed in our response to Reviewer #1, as well as, in the revised manuscript.

1. Regarding to Figure 1, the reviewer recommends the authors to repeat the same experiment at pH 7.4, where other processes of amyloid formation are operative. Such experiment provides insights into the nature of amyloid inhibition by crystallins, and improves the physiological relevance of the experiment.

Response: We thank the reviewer for the suggestion. In our research group we have previously contributed several important findings related to the aggregation mechanism of alpha-synuclein. In our study [Buell et al “Solution conditions determine the relative importance of nucleation and growth processes in alpha-synuclein aggregation”, 2014, PNAS, 111(21):7671-6], we have e.g. demonstrated that the aggregation mechanism of alpha-synuclein is pH dependent, where at neutral pH homogenous primary nucleation and secondary processes are undetectable. At this pH the rate of elongation and higher-order assembly of fibrils was found to be much higher. Further investigation resulted in [Gaspar et al, “Secondary nucleation of monomers on fibril surface dominates alpha-synuclein aggregation and provides autocatalytic amyloid amplification”, 2017, QRB-D, 50:e6], where we clearly demonstrated that at mildly acidic pH the overall mechanistic character of aggregation is changed, elucidating that the molecular mechanism underlying the aggregation process at mildly acidic pH is dominated by monomer-dependent secondary nucleation. 

Moreover, while neutral pH may add more physiological relevance to the study, it overlaps to some degree with what has been performed already by [Wauby et al in 2010, “The Interaction of αB-Crystallin with Mature α-Synuclein Amyloid Fibrils Inhibits Their Elongation”, 2010, Biophys J, 98(5): 843-851]. Noteworthy, and many times overlooked, is that acidic pH is found in some lumens of intracellular compartments, such as exosomes and lysosomes, making this pH range also very desirable in mimicking particular physiological conditions. 

Wauby et al have thus already successfully demonstrated that alphaB-crystallin inhibits elongation at neutral pH conditions [2010, Biophys J, 98(5): 843-851]. In addition to the physiological relevance of acidic pH, one of the main questions of our study is whether the crystallin protein family also has similar effects on secondary nucleation which is “switched on” at acidic pH. The reasoning behind such concerns is the fact that secondary nucleation consists of a double treat, rapidly increasing the load of fibrillar mass and smaller oligomeric species often linked to cytotoxicity. 

2. In the experiment shown in Figure S1, the authors examined the residual amount of aS monomer in the supernatant. This experiment is of importance, because several small compounds decrease ThT fluorescence without reducing the total amount of amyloid fibrils. However, two concerns arise regarding the experimental procedure. At first, it remains unclear whether the supernatant is truly devoid of aS amyloid fibrils (regarding to this point, centrifugation time and speed are not provided in the manuscript). I recommend the authors to measure ThT fluorescence before and after the centrifugation to examine what percentage of amyloid fibrils are precipitated by the centrifugation. Secondary, UV absorbance is not an accurate measure of protein concentration in a mixture containing two different proteins. Protein concentration should be determined by SDS-PAGE after separating aS from crystalins and measuring the band intensity.

Response: We thank the reviewer for this comment. We agree and have added the additional description of the method in the legend of the corresponding figure, which now includes the centrifugation speed and time. We acknowledge that although UV absorbance is not the ideal quantitative tool, it is a very nice approach and has been corroborated in parallel by complementary ex situ methods, as described in the paper [Arosio et al “On the lag phase in amyloid fibril formation” 2015, Phys Chem Chem Phys, 17(12): 7606-18]. This method was also proven useful in our own work [Grey et al, “Acceleration of alpha-synuclein aggregation by exosomes” 2014, J Biol Chem, 290(5): 2969-2982].

Concerning measuring ThT fluorescence before/after centrifugation, this would not lead to precise quantification as a result of low signal-to-noise ratio of ThT, failing to detect small concentration of aggregates as shown in [Arosio et al, “Quantification of the concentration of Abeta42 propagons during the lag phase by an amyloid chain reaction assay” 2014, JACS, 136(1): 219-225]. 

Finally, we fully agree that SDS-PAGE would be the ideal technique for quantification. However, it is still clearly evident from our data that the amount of fibrils decreases, which is the main message intended to convey with this additional supplementary experiment. 

3.The trap and seed kinetic experiment is tricky and less understandable. At first, it is difficult for me to understand which solutions marked by "filtrate 1", "control", "filtrate 2", etc., are filtrated once or twice. Furthermore, this experiment seems to contain a critical flaw: it was not proven that crystalins do not pass through the 200 nm filter. If not, crystalins will present in the trapped fraction at a high concentration, and inhibit generation of small oligomer in the next reaction. Crystalins form multimers, possibly larger than 200 nm. Furthermore, crystalins can interact with aS amyloid fibrils, which also potentially inhibit their flow-through.

Response: We have attempted to make the experimental description of the trap and seed assay clearer for the reader. All filtrates enumerated were filtered twice: once to trap the fibrils where the flow-through was discarded, and a second time to collect the monomer that had been incubated with the trapped aggregates which was supplemented with ThT and used to monitor aggregation. The control, the only condition that has been filtered only once, refers to the sample of monomeric alpha synuclein incubated alone with the filter plates used in this experiment. As alpha-synuclein is extremely sensitive to extrinsic factors, such as the surface material of the reservoir where it is studied, it is important to verify whether the surface material alone is capable of triggering aggregation, as is the case when using Polystyrene plates, reported in Gaspar et al, “Secondary nucleation of monomers on fibril surface dominates alpha-synuclein aggregation and provides autocatalytic amyloid amplification”, 2017, QRB-D, 50:e6. 

The idea of the trap and seed assay is exactly to trap crystallin-alpha synuclein complexes. What we observe is that both gamma-crystallin as well as alpha-crystallin pre-incubated with fibrils and subsequently trapped by filtration lead to a loss of the catalytic ability of such aggregates when incubated with freshly purified alpha-synuclein monomer. We can thus conclude that the inhibitory effect is related to an interaction between these two components, crystallin proteins and alpha-synuclein aggregates. This information is not directly extracted in the case where all components are mixed at time 0 (Figures 1 and 2). We can also exclude that (free) crystallins will be trapped as f.e. multimers in the filter. Their size, even in the case of the multisubunit alpha-crystallin (ca. 20nm in diameter), is well below the 200nm cutoff of the filter. We have added detailed information on the crystallin preparation and characterization of the sizes of the individual crystallins in the revised version of SI (see Figs S4-S7 and S2 and S3 Methods in SI) to make this clearer to the reader.

Changes in the revised manuscript: we have extended the description of the trap and seed experiments to make it clearer to the reader. The modified text now reads as follows:

Page 16-17: “α-Syn seeds were made from 40 µM monomeric α-syn under shaking conditions at 37ºC in Eppendorf tubes. Versatile GH-Polypro filter membranes of low-binding AcroPrep 96-well filter plates (Pall Life Sciences, Ann Arbor, MI) were initially washed with experimental buffer and saturated with monomeric α-syn. After, α-syn seeds with and without crystallin pre-treatment (2 h incubation) were trapped on these filter plates by filtration applying vacuum for 10 s and discarding the flow-through. The filtration was done using a MultiScreenHTS vacuum (Millipore) manifold. Monomeric α-syn was incubated with the trapped aggregates and newly filtered. This flow through was collected in 96-well non-binding PEGylated plates, supplemented with ThT and fluorescence intensity monitored in a plate reader at 37ºC under quiescent conditions. ”

4. In the abstract and discussion sections, the author hypothesized that crystallins might interact with aS fibrils. This hypothesize should be tested to provide insight into the nature of inhibition. For example, as described in doi: 10.1021/cn500019c, co-sedimentation assay is a simple method to detect interaction between amyloid fibrils and proteins. SPR analysis is alternative.

Response: We thank the reviewer for this very nice suggestion, which would indeed be very interesting. It is once again worth mentioning that Wauby et al in 2010 nicely showed alphaB-crystallin-alpha-synuclein complexes with immunoelectron microscopy [2010, Biophys J, 98(5): 843-851]. Moreover, and as outlined in our reply to question 3, we firmly believe that our trap and seed experiments strongly support our hypothesis of an interaction between crystallins and alpha-synuclein fibrils.

Reviewer #3: 

In this manuscript, Gaspar et al. investigate the effects of three types of crystallin proteins on the seeded amyloid fibril formation of alpha-synuclein at mildly acidic pH. The main finding is that the reaction of fibril formation is inhibited by these chaperone-like proteins. While it is not clear how relevant crystallins are for alpha-synuclein amyloid fibril formation in vivo, the study is nevertheless interesting and relevant. Not least because we still lack basic mechanistic understanding of alpha-synuclein amyloid formation under these conditions and any additional piece of information, also specific modes of action of inhibitors, is of value.

Overall I think the study is well-performed and the conclusions are mostly justified. However, I think that a few aspects need to be characterised/discussed in some more detail:

Response: We are very grateful to the reviewer for his/her positive feedback on the manuscript.

1. Some data needs to be shown from the purification of the proteins, such as SEC chromatograms, in order to give the reader a feel for how pure these preparations are, which are extracted from bovine eyes. Also, I would like to see a characterisation of the final chaperone preparations, at least in terms of their overall size distribution. Here, DLS, analytical ultracentrifugaion or microfluidic diffusional sizing would be appropriate techniques. Such measurements should be done both at pH 7, where the crystallins are extracted, as well as at pH 5.5, where the kinetic experiments are done.

Response: We thank the reviewer for pointing this out. We have accordingly complemented the revised SI with detailed information regarding the purification incl. chromatograms and characterization of crystallins using DLS at both pH 7 and 5.5. 

The respective section can be found on page 23 – 26 in SI.

2. Overall, the arguments that the crystallins inhibit secondary nucleation is supported by the data, in particular by the trap and seed experiments. However, the data also shows a different slope at time 0, which suggests an effect on elongation. This should be investigated in more detail, in order to be able to provide an estimate of the relative effects on elongation and secondary nucleation. Another noteworthy feature in the kinetic data is that the curves in the presence of inhibitor reach a plateau at lower ThT fluorescence intensities than the level in the absence of inhibtor. I guess for the amylofit analysis, the assumption was made that the ThT level is exactly proportional to the fibril concentration (this should be stated somewhere). That would mean that the inhibitor not only changes the kinetics, but also the final equilibrium position. This is interesting, as it should only happen with inhibitors that can interact with the monomer, and hence are able to shift the equilibrium. This seems not to be the case, however, for the crystallins. This should be commented on.

Response: We appreciate the very nice suggestions from the reviewer. To decompose the combined rate constants k+k2 into individual k+ and k2 would require many additional experiments. For k+, at least an estimation of fibril length from, for example, cryo-EM images would be required. The individual rates obtained in this manner would therefore also contain a source of uncertainty. Another downside is the fact that only one alpha synuclein monomer concentration was tested. 

Concerning the amylofit analysis, the information in the text has been updated and made clear.

On page 7, it now reads:

“The successful determination of such mechanistic insights rely heavily on the purity of the protein samples and on the reporter dye (ThT) in the sense that there is a linear correlation between signal and total aggregate mass.” 

To address the question related to the lower ThT plateau, we have also added on Page 5 the following:

“In the samples where aggregation occurs in the presence of α-crystallin, the ThT plateaus exhibit lower fluorescence intensities which suggests a decrease in the levels of ThT-positive α-syn aggregates. This may be a result of the inhibitory effect or an interference with the ThT signal.”

3. An estimate should be provided for the stoichimetry of fibril binding of the crystallins. For example one could incubate fibrils and crystallins and then spin them down and see how much of the crystallin is spund down together with the fibrils. This would be important to be able to compare its effect with that of other chaperones that can inhibit secondary nucleation in other amyloid systems, such as Abeta (e.g. Brichos chaperone). Also, there should be a somewhat more extensive discussion of the existing literature of secondary nucleation inhibitors in general and alpha-synuclein in particular, as some recent studies have presented efficient inhibition of alpha-synuclein secondary nucleation at highly sub-stoichiomtric ratio.

Response: We again thank the reviewer for the suggestion. A more extensive literature search has been performed and some relevant studies have been added in the discussion where it now reads on page 13: “Inhibition of secondary nucleation of α-syn was shown to occur by the homologous protein β-syn [47]. Several studies have also shown that substoichiometric concentrations of transthyretin (TTR) [48], Brichos [30], antibody fragments [49] and several small molecules [50] inhibit secondary nucleation of Aβ peptide”. 

While we have found many additional studies with small molecules including anle138b, BIOD303, fasudil, squalamine, EGCG, among others polyols, polyphenols and aromatic molecules, these were not included in the revised manuscript as we believe they fall outside the scope of the manuscript.

With respect to the stoichiometry experiment, we fear that if one were to spin down crystallin together with fibrils, free crystallin may also be dragged down leading to an overestimation. We believe from table 1 in the manuscript, it is evident from the minimal concentration that one observes inhibition, that the effect occurs far below equimolar concentrations, therefore very much in line with other recent reported substochiometric inhibitors of secondary nucleation.

On Page 11 of the revised manuscript we have thus added the following statement: “The observed extensive inhibitory effect of crystallin proteins far below equimolar amounts of α-syn is a first indication that their mechanism of action is not to interact with and reduce free α-syn monomer in solution but possibly to interact with the aggregated species affecting there catalytic nature.”,

---

## [Editor Report · Decision Letter 1]

11 Jun 2020

Eye lens crystallin proteins inhibit the autocatalytic amyloid amplification nature of mature α-synuclein fibrils

PONE-D-20-11144R1

Dear Dr. Ricardo Gaspar,

We’re pleased to inform you that your manuscript has been judged scientifically suitable for publication and will be formally accepted for publication once it meets all outstanding technical requirements.

Kind regards,

Peter Schuck

Academic Editor

PLOS ONE
---

## [Editor Report · Acceptance letter]

16 Jun 2020

PONE-D-20-11144R1 

Eye lens crystallin proteins inhibit the autocatalytic amyloid amplification nature of mature α-synuclein fibrils 

Dear Dr. Gaspar:

I'm pleased to inform you that your manuscript has been deemed suitable for publication in PLOS ONE. Congratulations! Your manuscript is now with our production department. 

Kind regards, 

on behalf of

Dr. Peter Schuck 

Academic Editor

PLOS ONE